# Unravelling three-dimensional adsorption geometries of PbSe nanocrystal monolayers at a liquid-air interface

Jaco J. Geuchies [1,2,4✉], Giuseppe Soligno [1], Ellenor Geraffy[1], Cedric P. Hendrikx[2], Carlo van Overbeek[1], Federico Montanarella[1], Marlou R. Slot [1], Oleg V. Konovalov[2], Andrei V. Petukhov [1,3] & Daniel Vanmaekelbergh[1✉]

The adsorption, self-organization and oriented attachment of PbSe nanocrystals (NCs) at liquid-air interfaces has led to remarkable nanocrystal superlattices with atomic order and a superimposed nanoscale geometry. Earlier studies examined the NC self-organization at the suspension/air interface with time-resolved in-situ X-ray scattering. Upon continuous evaporation of the solvent, the NC interfacial layer will finally contact the (ethylene glycol) liquid substrate on which the suspension was casted. In order to obtain structural information on the NC organization at this stage of the process, we examined the ethylene glycol/NC interface in detail for PbSe NCs of different sizes, combining in-situ grazing-incidence small-and-wide-angle X-ray scattering (GISAXS/GIWAXS), X-ray reflectivity (XRR) and analytical calculations of the adsorption geometry of these NCs. Here, we observe in-situ three characteristic adsorption geometries varying with the NC size. Based on the experimental evidence and simulations, we reveal fully three-dimensional arrangements of PbSe nanocrystals at the ethylene glycol-air interface with and without the presence of rest amounts of toluene.

[1] Condensed Matter and Interfaces & Physical and Colloid Chemistry, Debye Institute for Nanomaterials Science, Utrecht University, P.O. Box 80000, 3508 TA Utrecht, The Netherlands. [2] ESRF – The European Synchrotron, ID10, 71 Rue des Martyrs, 38000 Grenoble, France. [3] Laboratory of Physical Chemistry, Department of Chemical Engineering and Chemistry, Eindhoven University of Technology, P.O. Box 513, 5600 MB Eindhoven, Netherlands. [4] Present address: Optoelectronic Materials Section, Faculty of Applied Sciences, Delft University of Technology, Van der Maasweg 9, 2629 HZ Delft, The Netherlands. ✉email: j.j.geuchies@tudelft.nl; d.vanmaekelbergh@uu.nl

The self-assembly of semiconductor nanocrystals (NCs) on a liquid substrate has been pioneered by Murray et al.[1], who confined the formation of NC superlattices at a liquid–air interface. By drop casting a suspension of NCs in an apolar, volatile solvent on top of diethylene glycol, the NCs are forced to adsorb at the liquid–air interface and self-organize into large-area superlattices upon solvent evaporation. The exact superlattice structure that forms depends on interactions between the NCs[2,3] and the interaction of the particles with the interface. Recently, this method has been extended to form atomically connected NC solids through a process called oriented attachment: PbSe NCs adsorbed at the liquid/air interface align their atomic lattices and fuse via epitaxial {100}/{100} connections into an atomically coherent superlattice of one NC in thickness[4–6]. By tuning the synthesis conditions, various superlattice geometries could be obtained, such as a square superlattice geometry, where the NCs have a {100} facet pointing upwards, and a honeycomb super-lattice geometry, where the NCs have a {111} facet pointing upwards[6,7]. In both superlattice allotropes, the NCs attach via their {100} facets[8].

In situ synchrotron X-ray scattering techniques are nowadays being used more often to resolve the dynamics of self-assembly processes of lead chalcogenide NCs[9–12]. Recently, we[13] and others[14] have studied the formation mechanism of two-dimensional (2-D) square PbSe NC superlattices in situ and showed that the NCs undergo a remarkable sequence of phase transitions. Upon solvent evaporation, the NCs adsorb at the liquid–air interface and form a monolayer with hexagonally packed particles. Upon ligand detachment from the {100} facets, the NCs align crystal-lographically with a {100} facet pointing upwards. During this process, the hexagonal geometry of the superlattice is gradually changed to a square geometry. Finally, the particles attach epi-taxially via a necking process, where surface atoms move to form the connection between the NCs. There are still a large number of open questions regarding the described self-assembly process: how is the honeycomb superlattice formed? At which moment in the process is there a bifurcation toward either a square or a honeycomb geometry, and which factors decide the geometry? Which factors determine the amount of disorder on both atomic and NC length scales and how can we reduce this[10,15–21]? A recently introduced simulation model to predict the self-assembly of NCs at fluid–fluid interfaces provided fundamental insights that will help to answer these questions[22,23].

The adsorption geometry of the NCs at a liquid–air interface[24] should have a large impact on the final geometry of the NC superlattices. The pathway and the formation mechanism of silicene-like structures[7] with two distinctly different heights of two subsets of NCs remains a mystery. Ultimately, the NC interactions with the two liquids will determine the way the NCs adsorb at the interface[22,23,25]. Moreover, the interfacial adsorp-tion of the NCs will also create a capillary distortion of the liquid. Soligno et al. showed that cubes can create a hexapolar distortion of a liquid–liquid interface that could, in principle, induce directional "capillary" interactions between the NCs[26], but very probably these directional forces are too weak to dominate the assembly[22,23]. Recently, the relation between the surface chem-istry and the shape of PbSe NCs has been studied by Peters et al.[8]. They showed that chemisorption and surface reconfiguration results in a transformation of the NC shape from a truncated nanocube with rough surface facets to a truncated octahedron with larger and smooth {111} facets.

To study the three-dimensional (3-D) adsorption geometry of our NC monolayers, we perform three different X-ray scattering techniques quasi-simultaneously, which are shown schematically in Fig. 1. For the in situ grazing-incidence small-angle X-ray scattering (GISAXS) and grazing-incidence wide-angle X-ray

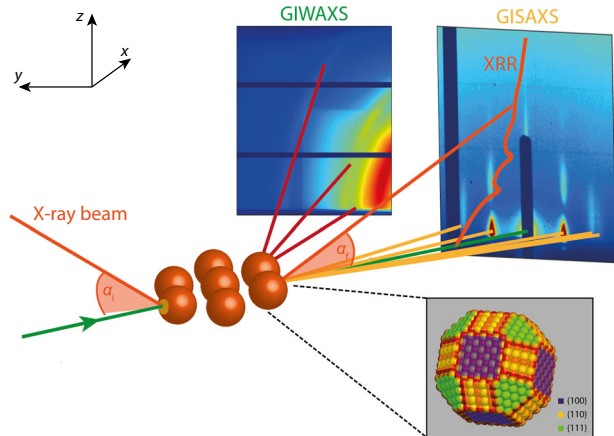

**Fig. 1 Schematic of the in situ GISAXS, GIWAXS, and XRR experiments performed at ID10 of the ESRF.** The GISAXS/GIWAXS experiments, which reveal the order of the NCs in the plane of the monolayer, are done using an angle of incidence of 0.14º with respect to the liquid surface. A detector is placed in the forward scattering direction to collect the GISAXS data, which reveals information on the inter-nanocrystal order. A second detector is placed at a higher angle and closer to the sample to collect the GIWAXS signal, which reveals information on the crystallographic orientation of the NCs with respect to the NC monolayer. Upon completion of solvent evaporation, the specular XRR is collected using a dual crystal deflection scheme; the angle of incidence is varied and the intensity of the specular reflection is recorded. The inset shows a schematic of a PbSe truncated nanocube.

scattering (GIWAXS) experiments, the incoming X-ray beam glances the liquid–air interface at an incident angle, $\alpha_i$, of 0.14º, i.e., the critical angle for total external reflection of the X-ray photons at 22 keV for PbSe. The GISAXS pattern is recorded in the forward direction and reveals information on the periodicity and order in the NC monolayer. The atomic diffraction is recorded at a detector placed closer to the sample under a higher angle. The collected GIWAXS signal allows us to obtain the crystallographic orientation of the NCs at the liquid–air interface. The before mentioned techniques are complemented with spec-ular X-ray reflectivity (XRR) measurements. Using the double-crystal deflection diffractometer at the ID10 beamline of the European Synchrotron Radiation Facility (ESRF), the angle of incidence is varied and the intensity of the specular beam is recorded on a one-dimensional (1-D) detector on the dif-fractometer arm. The scattering vector for this specular reflection only has a component in the vertical direction, i.e., $q_z$, which allows us to obtain information on the average density profile of the NC monolayer in the direction perpendicular to the liquid–air interface. When a monolayer of NCs is present at the liquid–air interface, the signal is modulated owing to constructive and destructive interference upon scattering and so-called Kiessig fringes can be observed. These fringes modulate with a period $2\pi/\Delta$ in reciprocal space, where $\Delta$ is the thickness of the densified colloidal system formed at the interface, for instance, a NC monolayer[27].

The focus of this work is on the nearly last stage of the process of PbSe superlattice formation, i.e., when the NC monolayer rests on the ethylene glycol (EG) substrate, possibly before atomic attachment has taken place. We adsorb PbSe NCs with different sizes at the EG–air interface and study the in-plane NC geometry with GISAXS and their crystallographic orientation with GIWAXS. Furthermore, we extend these two techniques with specular XRR measurements[28,29], which allowed us to obtain the density profiles of the NC monolayers in the direction

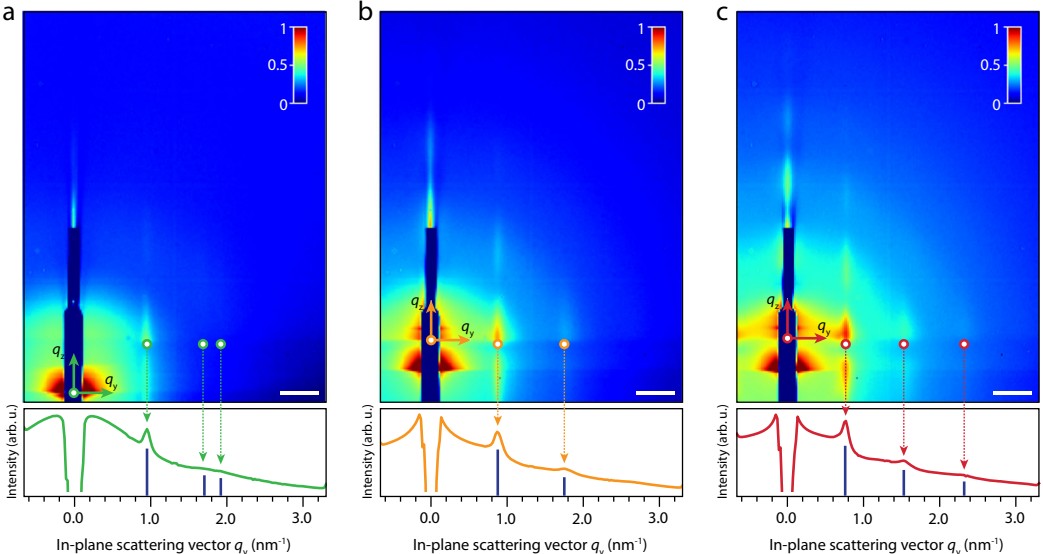

**Fig. 2 Representative GISAXS patterns of small-, intermediate-, and large-sized PbSe NCs at the ethylene glycol–air interface.** Intensity traces in the horizontal scattering direction $q_y$ are depicted below the GISAXS patterns. The obtained Bragg rod positions are indicated with blue vertical lines below the intensity traces. **a** GISAXS pattern of the small NCs. The relative positions of the Bragg rods in the horizontal scattering direction (depicted with green arrows) indicate the formation of a hexagonally close packed monolayer of NCs. **b** GISAXS pattern of the medium-sized NCs. The position of the Bragg rods (depicted with yellow arrows) indicates that there is a preference for NCs ordering in one dimension. **c** GISAXS pattern of the large NCs. The relative positions of the Bragg rods again indicate that there is a preference for NC ordering in one dimension. Comparing the small-, medium-, and large-sized NCs, the Bragg rods move inward, indicating an increased center-to-center distance of the particles in the NC monolayer. Scale bars in the GISAXS patterns equal 0.5 nm$^{-1}$.

perpendicular to the EG/air interface. The data presented here is (also) an important step forward in the understanding the formation mechanism of 2-D superlattices, but the main goal of this work goes beyond this: we unravel the 3-D adsorption profile of PbSe NCs at fluid–fluid interfaces using a unique and novel combination of different experimental and numerical techniques.

## Results and discussion
**In situ synchrotron X-ray scattering.** We synthesized NCs with varying sizes in the range of 4–10 nm, as outlined in Supplementary Methods. From the literature, it is already known that the truncation of the NCs is decreased when increasing the NC size[30]. The size of the {100} facets increases at the expense of the {110} and {111} facets giving more cubic-shaped NCs. We divided the used NCs in three size ranges: small-sized NCs with a diameter <5.5 nm, medium-sized NCs with diameter in the range 5.5–7.6 nm, large-sized NCs with diameter in the range 8.2–9.1 nm. Transmission electron microscopic (TEM) images and absorption spectra of the synthesized NCs can be found in Supplementary Methods (see Supplementary Table 1 and Supplementary Figures 1-3 for a summary of the NCs used throughout this study).

The NC dispersion in toluene is dropcasted inside the liquid cell on EG, which acts as an immiscible liquid substrate. To the EG substrate, we add 100 μL of a 31.7 μM oleic acid solution in EG solution for two reasons: first, to increase the wettability of the toluene droplet with NCs on the EG. The second reason is that addition of oleic acid to the EG sub-phase has proven to stop oriented attachment of the NCs[4]. The single NC adsorption geometry is key to understand the behavior of these NCs at the EG–air interface, which is why we attempt to block the epitaxial fusing of the NC monolayers.

First, we present and discuss GISAXS data, see Fig. 2. The scattering patterns are acquired after 2 h of solvent evaporation from the casted NC suspensions, so that most toluene should be evaporated from the NC dispersion. The GISAXS signal for the

small NCs, shown in Fig. 2a, shows Bragg rods at positions of 0.94 nm$^{-1}$, 1.62 nm$^{-1}$, and 1.88 nm$^{-1}$ in the horizontal scattering direction $q_y$ with a full width at half maximum (FWHM) of 0.06 nm$^{-1}$. The peak positions of 1:√3:2 indicate that the NCs are ordered in a 2-D hexagonal lattice with an NC center-to-center distance of 7.7 ± 0.4 nm, roughly the NC diameter plus interdigitated oleic acid ligands.

The GISAXS pattern of the medium-sized NCs, depicted in Fig. 2b, shows Bragg rods at 0.86 nm$^{-1}$ and 1.71 nm$^{-1}$ in the $q_y$ direction with an FWHM of 0.08 nm$^{-1}$. The 1:2 relative peak positions indicate that there is a preference for NC ordering in one dimension, e.g., the formation of linear structures. The NC center-to-center distance is calculated to be 7.3 ± 0.7 nm.

The relatively small center-to-center distance of the NCs compared to the NC diameter is not fully understood. Hanrath and co-workers showed through molecular dynamics simulations that interparticle distances as small as 0.5 nm can be achieved when the ligand density of oleic acid on PbSe is low enough[31]. This is a hard parameter to quantify during these experiments, as the ligand density on the NC surface likely changes throughout the self-assembly process. For the large NCs, the GISAXS signal depicted in Fig. 2c shows Bragg rods at 0.75 nm$^{-1}$, 1.52 nm$^{-1}$, and 2.18 nm$^{-1}$ in the $q_y$ direction with an FWHM of 0.05 nm$^{-1}$. Again the 1:2:3 relative peak positions indicate that there is a preference for NC ordering in one dimension. The center-to-center distance is calculated to be 8.4 ± 0.5 nm. This is smaller than the center-to-center distance expected for two NCs with two layers of oleic acid between them (length oleic acid ~1.8 nm[32]). Further discussion follows below. The GISAXS experiments do not imply that we are monitoring linear structures, as a small degree of disorder and deviation of a 90° (square NC ordering) or 60° (hexagonal NC ordering) bond angle removes the in-plane correlation peaks[13]. The occurrence of oriented attachment cannot be excluded, which would explain the decreased center-to-center distances of the NCs, which will be discussed further in the sections on the GIWAXS and XRR data. The full GISAXS

dataset on all NC sizes can be found in Supplementary Information (Supplementary Figs. 4–6, a summarized version is presented in Supplementary Table 2).

The crystallographic orientation of the NCs is obtained by measuring the diffraction from their atomic lattices, which is measured simultaneously with the GISAXS data and is presented in Fig. 3. We calculated the expected GIWAXS pattern for NCs having a [001] axis pointing perpendicular to the liquid–air interface in Fig. 3a. Figure 3b–d show the GIWAXS patterns corresponding to the small-, medium-, and large-sized NCs, respectively. For the small NCs, we observe powder rings in the GIWAXS pattern, which means that the NCs do not have a preferential orientation at the liquid–air interface. Either the NCs are still freely rotatable or the ensemble of NCs have random, static orientations, which will average out in a ring in the GIWAXS pattern.

The GIWAXS pattern of the medium-sized NCs, shown in Fig. 3c, shows a series of well-defined diffraction spots. An azimuthal intensity trace over the reflection originating from the {222} planes shows that it has an orientation of 36.0° with respect to the liquid–air interface, corresponding to NCs having a {001} facet pointing upwards. The FWHM of the reflection, which is an indication of the degree of rotational freedom of the <100> axis perpendicular to the liquid surface the NCs still have at the liquid–air interface, is 6.5°. The GIWAXS pattern of the large NCs, shown in Fig. 3d, matches the calculated GIWAXS pattern very well. It shows that the NCs have the same orientation as the medium-sized NCs, i.e., a {001} facet pointing upwards. The variation in orientations is slightly smaller, as the FWHM of the 222 reflection is 6.1°. This means that the large NCs have less rotational freedom at the liquid–air interface. We show the width of the Bragg peaks along the azimuthal directions for the 222, 420, 422, and 311 reflections versus the NC diameter in Supplementary Table 3. The apparent decrease in peak width with increasing NC size indicates an increase in crystallographic alignment of the NCs. The GIWAXS patterns of all NCs are presented in Supplementary Figures 7-10.

The orientation of the NCs is further verified by looking at intensity traces in the 2θ direction along the scattering horizon (azimuthal angle ~1°), which we presented for the same samples in Supplementary Information (Supplementary Fig. 19). For the medium- and large-sized NCs, only reflections originating from {hk0} PbSe lattice planes are observed along the scattering horizon. These particular atomic planes are oriented perpendicular to the liquid–air interface when the NCs have a [001] axis pointing upwards and hence scatter horizontally. Previously observed superlattices from PbSe NCs have been shown to show attachment of their {100} facets[4,7]. We are able to verify whether or not the NCs are attached by looking at the FWHM of the 400 reflection in the horizontal scattering direction and estimating the crystalline size of the NCs. The results are presented in Table S10. Since the diameters obtained using the Scherrer equation are not significantly larger than the NC diameters determined by TEM, attachment does not occur. Possibly, there are some regions on the sample with and without attached NCs. We have also taken this into account in the analysis of the reflectivity data, which will be discussed below.

NC alignment was also observed by Van der Stam et al., who recently showed that, upon addition of oleic acid, 11 nm polyhedral ZnS bifrustum NCs align atomically with their {002} facet pointing upwards at the EG–air interface[33,34]. Hanrath et al. showed that large cubic PbSe NCs with an edge length of 25 nm do align a [111] axis perpendicular to the toluene–air interface, but there the NCs formed 3-D body-centered cubic super-lattices[25]. This is also the orientation that PbSe NCs are required to have before silicene-type honeycomb superlattices can

form[7,22]. We will show in the section below that the full adsorption geometry of the PbSe NCs studied here is determined by an interplay between the adsorption energy, mostly dictated by the NC size, and the degree of truncation of the NCs.

To obtain quantitative information in the direction perpendicular to the liquid–air interface, we performed specular XRR measurements. When one measures the specular reflection, the scattering vector **q** only has a component perpendicular to the interface z. The interference of the X-ray photons reflected from a stratified surface will give rise to periodic intensity oscillations, the Kiessig fringes. The periodicity of the oscillations contains information on the layer thickness, whereas the scattered intensity depends on the averaged scattering density profile across the interface. This means that one can fit the acquired reflectivity curve to get detailed information on a materials density gradient in the z-direction, e.g., how NCs adsorb at liquid–air or liquid–liquid interfaces.

The adsorption behavior of NCs at interfaces has already been studied using a combination of X-ray techniques. Vorobiev et al. studied self-assembly of iron oxide nanoparticles of different sizes at the water–air interface under different surface pressures to show the optimal conditions for making monolayer films of the particles[35]. XRR has been used recently in a combination with GISAXS to study various types of Au nanoparticles on different substrates[36,37]. Kosif et al. were able to produce rigid NC films at the water–air interface by connecting gold nanoparticles together with thiol-group containing linker molecules[38]. They studied the film structure under different surface pressures using grazing-incidence X-ray diffraction and XRR to see how much force was necessary to buckle the NC film.

We continue the discussion here with a comparison of the XRR data from the small- and medium-to-large-sized PbSe NCs presented in Fig. 4 (the full set of XRR measurements are presented in Supplementary Figure 11-19, and fitted parameters are displayed in Supplementary Tables 4-9). For the fitting of the data, we apply a recursive fitting procedure based on a Parratt formalism[39]. The fit takes into account the position and orientation of the particle with respect to the liquid–air interface, the degree of truncation of the NCs, the thickness of the ligand corona around the NC, and several density and roughness parameters (see Supplementary Information for an extensive discussion on the XRR data analysis). We observe three characteristic density profiles perpendicular to the EG–air interface, which can be categorized into those for (1) small-sized NCs, (2) large- and medium-sized NC monolayers, and (3) large- and medium-sized NC monolayers, where a certain degree of buckling is observed; sometimes, one NC is displaced downwards by roughly half an NC diameter. The observation of these buckled layers is striking, since their presence is not readily deduced from the GISAXS data, which shows the necessity of the specular XRR measurements to fully characterize the adsorption behavior of these NCs.

Figure 4a shows a representative XRR curve from the small NCs as blue dots, with the corresponding fit as a red solid line. The clear oscillations of the signal only show one period, proving that we truly are looking at a monolayer of NCs. The value of the real part of the refractive index of each layer $j$, $\delta_j$, is proportional to the electron density of that layer. This is plotted in Fig. 4d as the blue curve; the yellow curve is the first derivative of this density profile, which we use to identify the position of the particle with respect to the liquid–air interface. The middle of the scattering length density (SLD) profile corresponds to a mean value for the center of mass of the NCs above the liquid–air interface level, which lies 3.8 nm above the EG–air interface, i.e., the NCs do not penetrate into the EG. This can be rationalized by speculating that toluene is still

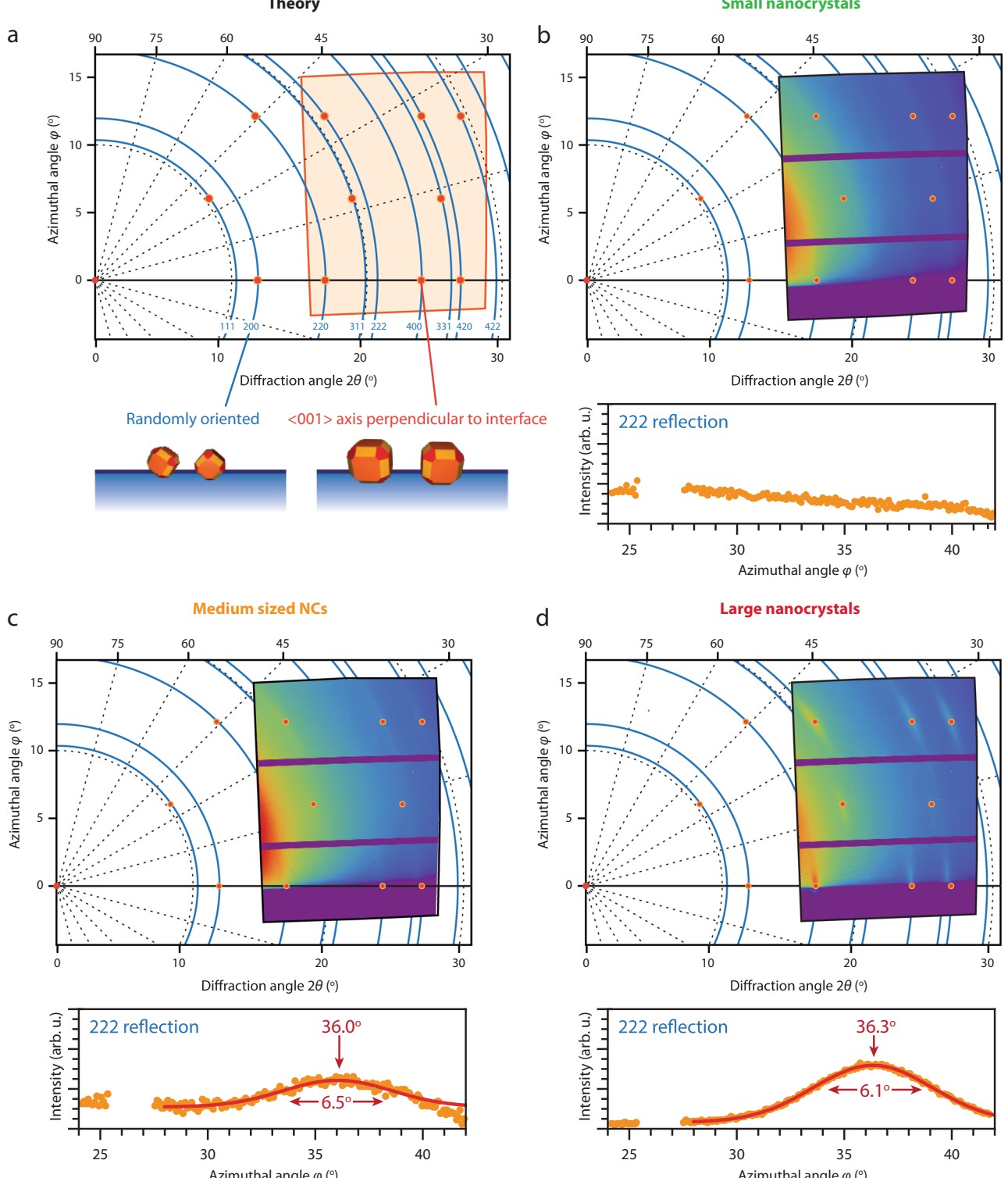

**Fig. 3 Analysis of the corresponding GIWAXS patterns reveal crystallographic alignment of larger PbSe NCs with respect to the liquid–air interface.**
Intensity traces are taken over the 222 diffraction ring versus their azimuthal angle $\varphi$ and shown below the image. **a** Calculated GIWAXS pattern for rocksalt PbSe NCs having a [001] axis perpendicular to the liquid–air interface (red dots) and when allowed full rotational freedom (blue rings). The orange area shows the position of the GIWAXS detector in this coordinate space. **b** GIWAXS pattern of the small NCs showing no preferential crystallographic alignment. **c** GIWAXS pattern of the medium-sized NCs, which can be modeled well to NCs pointing a [001] axis upwards, perpendicular to the liquid–air interface. The width of the 222 reflection is an indication of how well the NCs are atomically aligned with respect to the liquid–air interface. The medium-sized nanocrystals have a 6.5° variation in orientation (FWHM). **d** GIWAXS pattern of the large NCs, which can also be modeled well to NCs pointing a [001] axis upwards. The width of the 222 reflection is smaller compared to the same reflection of the medium-sized NCs; the large nanocrystals have a 6.1° variation in orientation (FWHM).

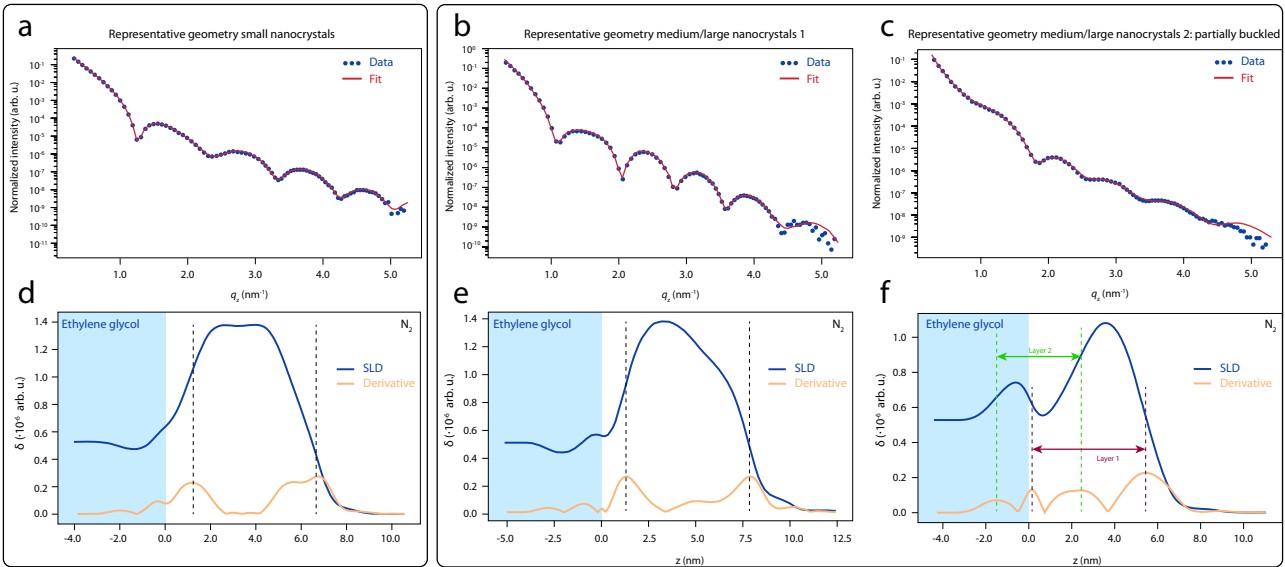

**Fig. 4 Specular XRR reveals three characteristic density profiles perpendicular to the ethylene glycol–air interface. a** Representative XRR curve from a monolayer of small PbSe NCs. The red line is the best fit of the data, in which we approximate the NCs as truncated nanocubes. **b** Representative XRR curve from a monolayer of medium-/large-sized PbSe NCs. The red line is the best fit of the data. **c** For large and medium-sized PbSe NCs, we observe a second type of adsorption geometry, where the nanocrystals slightly buckled, i.e., two layers on top of each other. **d** SLD plot from the fit for the small NCs displayed in **a**. The blue curve represents the density profile; the yellow curve is the first derivative of this density profile. The nanocrystal monolayer floats on top of the ethylene glycol. **e** SLD plot from the fit displayed for the medium-/large-sized NCs displayed in **b**. Again, the NC monolayer floats on top of the ethylene glycol. **f** SLD plot from the fit for the second adsorption geometry observed for medium- and large-sized PbSe NCs displayed in **c**. As can be seen, there are two NC layers, which are displaced by roughly half a particle diameter from each other in the direction perpendicular to the ethylene glycol–air interface.

adsorbed in the NC ligand corona (a more detailed argument follows in the theory section).

Figure 4b presents a representative XRR curve from the medium-/large-sized NCs as blue dots, with the corresponding fit as a red solid line. Again, only one periodicity of the XRR signal is observed, confirming that we are measuring a NC monolayer again. The corresponding SLD plot, shown in Fig. 4d, shows that the center of mass of the large NC monolayer is sticking out 3.0 nm above the EG–air interface. This is one out of two typically observed adsorption geometries for the medium/large sized NCs. This also explains the easy transfer to solid substrates using a Langmuir–Schaefer-type stamping technique. In Fig. 4c, we show the XRR curve of the second typically observed adsorption geometry, which we roughly observed in 20% of our experiments on the medium-/large-sized NCs. The density profile is displayed in Fig. 4f. Two NCs layers are observed in the SLD profile, where one layer is displaced by roughly half an NC diameter downwards with respect to the EG–air interface. This observed "buckling" of the NC monolayers resembles the expected buckling of the honeycomb superlattice[5,7], with the exception that these NCs are oriented with a {100} facet pointing upwards from the liquid–air interface. Here there is truly the penetration of some NCs through the interface. We note that the observed NC adsorption geometry could differ strongly from the initial adsorption geometry at the toluene–air interface. Future in situ experiments will be focused on obtaining a stable (i.e., non-evaporating) toluene–air interface to see how the NCs adsorb and align during the early stages of the self-assembly process.

**Analytical calculations on the NC adsorption geometry.** We corroborate and clarify the presented experimental data with analytical calculations to predict the equilibrium adsorption geometry of the PbSe NCs at the toluene/air and EG/air interface. An internal energy $\gamma A$ is associated with a fluid–fluid interface of

surface area $A$, with $\gamma$ the fluid–fluid surface tension. A micro-particle or nanoparticle can reduce the internal energy of the system by adsorbing at the fluid–fluid interface, since this reduces $A$. Treating the two fluids as homogeneous, the total internal energy associated with a particle staying at the interface between the two fluids (say fluid 1 and fluid 2) is[40,41]

$$U = \gamma A + \gamma_1 A_1 + \gamma_2 A_2, \qquad (1)$$

where $A_1$ and $A_2$ are the surface areas of the particle in contact with fluid 1 and fluid 2, respectively, and $\gamma_1$ and $\gamma_2$ is the surface tension of the particle surface with fluid 1 and fluid 2, respectively. In Eq. (1), we assume that the fluid–fluid interface is flat far away from the particle, so the fluid pressure–volume terms can be neglected in $U$ (see Supplementary Information). Gravity is not included in Eq. (1), since this is negligible for nanoparticles compared to the surface energy terms.

At equilibrium, the particle stays at the interface with the position and orientation that minimize $U$[40]. Such a minimum-$U$ orientation is fundamental for directing the self-assembly of NCs at fluid–fluid interfaces[22]. To improve our understanding of the self-assembly of PbSe NCs in honeycomb and square superstructures[4,6,7,13], we theoretically predict the equilibrium orientation of a single PbSe NC at the various fluid–fluid interfaces involved in the self-assembly experiments, that is toluene–air and EG–air. In principle, also the EG–toluene interface is involved in the experiments, but the NCs are not expected to adsorb at this interface (see Supplementary Information).

We numerically compute $U$ for various orientations $\varphi$, $\psi$ of a NC at a flat fluid–fluid interface, where the $z = 0$ plane corresponds to the interface without NC, $\varphi$ is the polar angle of the NC vertical axis with respect to the $z$-axis, and $\psi$ is the Euler internal angle of the NC around its vertical axis, as shown in Fig. 5a. As NC surface, we consider both a rhombicuboctahedron and a cantellated rhombicuboctahedron (the exact definition

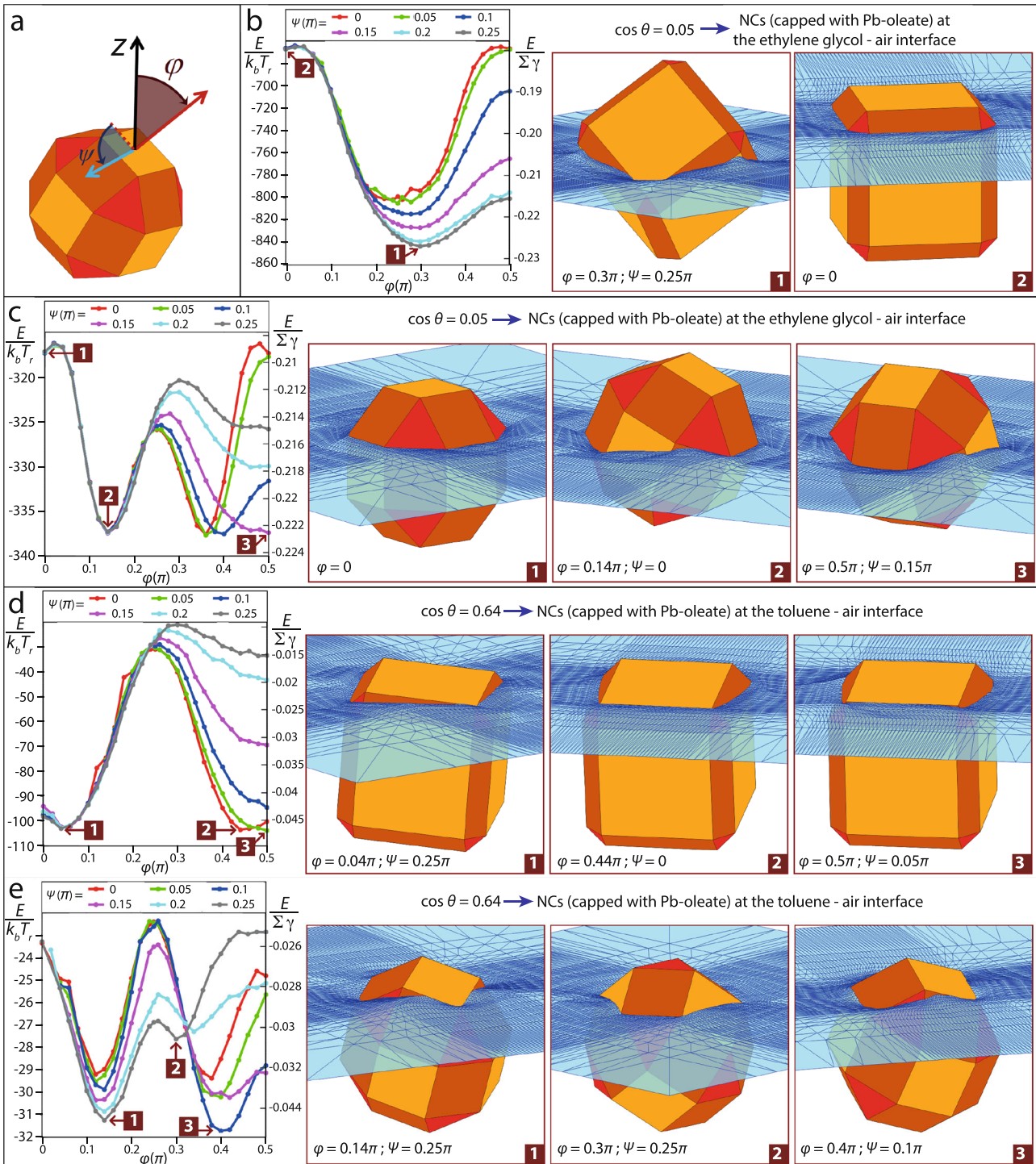

**Fig. 5 Calculated equilibrium adsorption geometries of PbSe NCs with different degrees of truncation. a** Orientation $\varphi$, $\psi$ of an NC at a flat fluid–fluid interface (parallel to the plane $z = 0$), where $\varphi = \in [0, \pi]$ is the polar angle of the NC vertical axis (red arrow) with respect to the $z$-axis, and $\psi = \in [0, 2\pi]$ is the internal Euler angle of the NC around its vertical axis (with $\psi = 0$ such that the NC has a {100} facet parallel to $z = 0$ when $\varphi = 0.5\pi$. **b–d** Energy $E(\varphi, \psi)$ [Eq. 2], as computed by the numerical method by Soligno et al., for an NC with **c, e** rhombicuboctahedron and **b, d** cantellated rhombicuboctahedron shape. The Young's contact angle is **b, c** $\cos(\theta) = 0.05$ and **d, e** $\cos(\theta) = 0.64$, corresponding to an NC (capped with oleate ligands) at the ethylene glycol/air and toluene/air interface, respectively. $E$ is shown both in units of $\Sigma\gamma$ (with $\Sigma$ the NC total surface area and $\gamma$ the fluid–fluid surface tension) and in units of $k_b T_r$ (thermal energy at room temperature). For **b, c**, $\gamma = 0.047$ N/m (EG/air surface tension) and **d, e** $\gamma = 0.028$ N/m (toluene/air surface tension) and for an NC with size **b, d** 8 nm ($\Sigma \cong 325$ nm$^2$) and **c, e** 6 nm ($\Sigma \cong 133$ nm$^2$). For symmetry reasons, we show $E$ only for $\varphi = \in [0, \pi/2]$ and $\psi = \in [0, \pi/4]$, since other NC orientations are equivalent. Note that $E$ is automatically minimized with respect to the height of the NC center of mass on the interface level far away from the NC (see Supplementary Information for more details). The insets show a 3-D view, close to the NC, of the equilibrium shape of the fluid–fluid interface (blue grid, with the fluid on the top being air), as computed with Soligno's method, for the predicted equilibrium and metastable $\varphi$, $\psi$ orientations of the NC (indicated by the labels). In **c**, some of the NC metastable orientations are not shown due to the limited space of the figure.

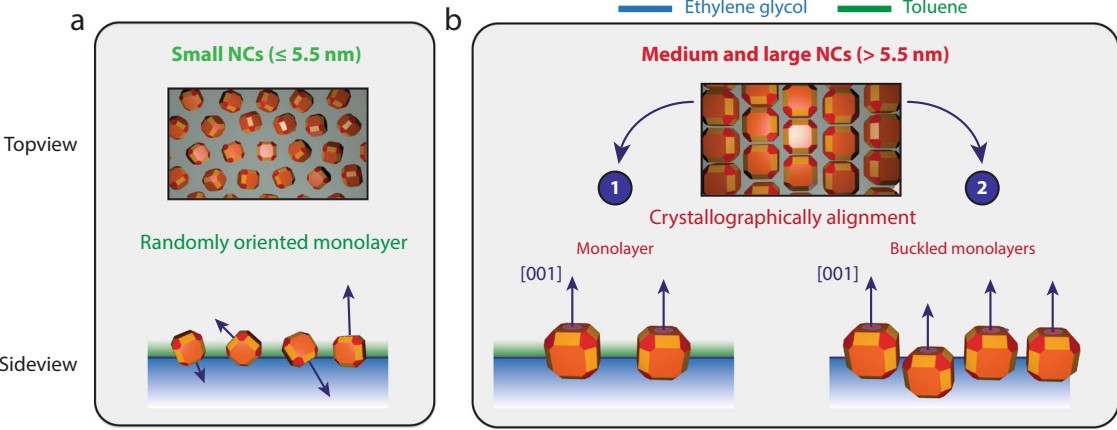

**Fig. 6 Schematic representation of the NCs adsorbed at the ethylene glycol–air interface.** Combining the information from the GISAXS data (NC–NC distance), the GIWAXS data (crystallographic orientation), and the XRR measurements (height of the NCs with respect to the interface), we can create quasi 3-D models of how the PbSe NCs adsorb at the ethylene glycol–(toluene)–air interface. **a** The small NCs are randomly oriented at the liquid–air interface, presumably due to the limited faceting of the particles. **b** The medium-sized and larger NCs attain a more cubic shape, with larger {100} facets compared to the smaller NCs, making it favorable to crystallographically align their atomic lattices at the liquid–air interface with a (100) facet pointing upwards, as confirmed by the theoretical predictions in Fig. 5. There are two possible adsorption geometries for these NC sizes as extracted from the XRR measurements: (1) a monolayer that floats on top of the ethylene glycol–toluene–air interface, and (2) a buckled layer, where one NC layer floats on top of the ethylene glycol, and some NCs penetrate into the ethylene glycol.

is given in Supplementary Information). These shapes are approximations meant to capture the key geometrical features of PbSe NCs, which typically have a highly truncated cubic shape when their size is around or <6 nm, and a slightly truncated cubic shape for larger sizes. For each considered orientation $\varphi$, $\psi$ of the NC, the unknowns $A$, $A_1$, $A_2$ in Eq. (1) are computed with the numerical method introduced by Soligno et al.[40–42], where a simulated annealing algorithm is used to find the equilibrium shape of the fluid–fluid interface for the given NC orientation (see Supplementary Information for technical details on our calculations). Our approach ensures that the effects due to the capillary deformations induced by the adsorbed NC in the equilibrium shape of the fluid–fluid interface are included in $U$. As shown in ref. [17], neglecting capillary deformations (the so-called Pieranski approximation[34,43–45], corresponding to assuming that the fluid–fluid interface is flat everywhere also in the presence of the NC) can lead to qualitative errors in the predictions for the NC equilibrium orientation.

Our theoretical results are presented in Fig. 5. For convenience, we show $E(\varphi, \psi)$ instead of $U(\varphi, \psi)$, where

$$E \equiv \gamma (A - A_0 + A_1 \cos \theta) \qquad (2)$$

is just $U$ shifted by a constant (see Supplementary Information), such that $E = 0$ corresponds to the NC desorbed from the interface and fully immersed in fluid 2. In Eq. (2), $A_0$ is the area of the fluid–fluid interface when no NC is present, and the parameter $\theta$ is Young's contact angle, defined by Young's Law, that is[34]

$$\cos \theta = (\gamma_1 - \gamma_2)/\gamma \qquad (3)$$

To represent a PbSe NC (capped by oleic acid ligands) at an EG/air and toluene/air interface, we set $\cos\theta = 0.05$ and $\cos\theta = 0.64$, respectively (see in Supplementary Information a justification for these numbers). In Fig. 5, $E$ is shown both in units of $\Sigma\gamma$ (with $\Sigma$ the NC total surface area) and, for a given value of the NC size and of $\gamma$, in units of $k_B T_r$ (with $k_B$ the Boltzmann constant and $T_r$ room temperature).

A highly truncated cubic NC with size 6 nm, see Fig. 5c, e, is bonded by an energy well $\sim -340\, k_B T_r$ at the EG/air interface and $\sim -30\, k_B T_r$ at the toluene/air interface. Therefore, the NC prefers

to adsorb at the EG/air interface. However, in the experiments, we expect the NC to first adsorb at the toluene/air interface, since the EG/air interface does not exist until all the toluene around an NC is evaporated (and the NCs do not adsorb at the EG/toluene interface, see Supplementary Information). Once the NC is at the toluene/air interface, it remains there until the toluene is completely evaporated, since the energy barrier to spontaneously desorb from the toluene/air interface ($\sim 30\, k_B T_r$) is too high at room temperature. The SLD plots of Fig. 4d, e show that the small- and medium-sized NCs did not adsorb in the EG phase. We hypothesize that a thin layer of toluene was still on top of the EG phase when the X-ray scattering measurements were done, such that the NCs were confined at the toluene–air interface (or at least, toluene was still present in the ligand corona of the NCs). Indeed, if all the toluene was evaporated, then the NCs should have been adsorbed at the EG/air interface and be half-immersed in the EG phase and half-immersed in the air, as shown in the insets of Fig. 5c, which is in contrast with the experimental results, see Fig. 4d, e. Possibly, the evaporation of the last amount of toluene from inside the ligand corona of the NCs is much slower than the evaporation of the bulk toluene liquid.

Figure 5e shows that the 6-nm highly truncated NC has multiple metastable orientations at the toluene/air interface, separated by energy barriers of a few $k_B T_r$, suggesting that the NC has essentially random orientation at this interface, in agreement with the experimental results in Fig. 3b, c.

A slightly truncated cubic NC with size 8 nm, see Fig. 5b, d, is bonded by an energy well $\sim -840\, k_B T_r$ at the EG/air interface and $\sim -100\, k_B T_r$ at the toluene/air interface. Therefore, for the same argument previously illustrated, we expect the NCs to remain at the toluene/air interface as long as the toluene is not evaporated. The SLD plots of Fig. 4f show that some of the large-sized NCs are adsorbed in the EG phase, while others remain on top of it. This suggests that toluene was still present close to the NCs on top of the EG phase, while it was essentially evaporated around the NCs that managed to adsorb in the EG phase.

The larger-sized NCs experimentally are found oriented with a {100} facet parallel to the interface plane, see Fig. 3d. For the NCs still adsorbed at the toluene–air interface, i.e., not immersed in the EG phase, this orientation matches with our predictions for a

slightly truncated cubic NC at a toluene–air interface, see Fig. 5d. For the NCs partially immersed in the EG phase, so staying at the EG–air interface, we would expect as equilibrium orientation a {111} facet parallel to the interface plane, see Fig. 5b. However, a metastable orientation with a {100} facet parallel to the interface is also predicted for a slightly truncated NC at the EG–air interface, see Fig. 5b, with energy barrier ~5 $k_B T_r$. This seems to explain why the larger-sized NC adsorbed in the EG phase are also found oriented with a {100} facet parallel to the interface: since this is their orientation when they are adsorbed at the toluene–air interface, they remain trapped in the same orientation when the toluene completely evaporates and they adsorb at the EG/air interface.

Combining the data from the GISAXS, GIWAXS, and XRR experiments, and strengthening them with our calculations, we can create a size- and shape-dependent 3-D model of how PbSe NCs adsorb at the EG–(toluene)–air interface. A schematic representation is shown in Fig. 6. The smaller NCs (with sizes ≤5.5 nm) are oriented randomly, which results from the smaller adsorption energy and the reduced size of the NC facets compared to larger NCs. As the NCs grow in size, their adsorption energy is increased, the truncation of the particles is reduced, and they get a more cubic shape with larger {100} facets. The delicate interplay between adsorption energy, which is directly related to the NC size, and truncation parameter of the NCs gives the specific adsorption geometries that we determined experimentally.

Future X-ray scattering experiments should focus on different aspects of the self-assembly process: (1) Varying the ligand density and different ratios of facet sizes, which are intimately related, to identify key parameters driving the formation of either honeycomb or square 2-D superlattices. (2) Unraveling the adsorption geometry of the PbSe NCs at the toluene–air interface. These experiments are more difficult, as the evaporation of toluene has to be stopped to perform the reflectivity measurements. A solution could lie in using specially developed liquid cells, which controllably saturate the atmosphere inside the cell with solvent vapor[46].

To summarize, we studied the adsorption behavior of PbSe NCs at the EG–air interface with and without the presence of residual amounts of toluene, using a combination of X-ray scattering techniques. Furthermore, we combine GISAXS and GIWAXS with specular XRR measurements to obtain full 3-D pictures of how these NCs adsorb at the EG–(toluene)–air interface. We show that larger PbSe NCs align crystallographically with a [001] axis perpendicular to the liquid–air interface, due to an interplay between adsorption energy and the degree of truncation of the NCs. These experiments were corroborated with analytical calculations of the NC equilibrium adsorption geometry to rationalize the NC behavior at the various fluid–fluid interfaces as a function of NC size and truncation. The adsorption geometry of the NCs in the early stages of oriented attachment are expected to have great impact on the atomically connected 2-D superlattices. The experiments presented throughout this work show that it is possible to reveal 3-D arrangements of NC monolayers adsorbed at liquid–air interfaces, from the NC to atomic length scales. This particular combination of techniques increases our understanding of self-assembly processes on liquid substrates and will help guide the fabrication of novel 2-D superstructures.

## Methods

**NC synthesis**. The PbSe NCs used for the oriented attachment experiments in this study were prepared using an adapted method described by Steckel et al.[47]. Details can be found in Supplementary Methods.

**In situ synchrotron X-ray scattering**. The in situ X-ray scattering experiments under grazing incidence were performed at beamline ID10 of the ESRF, Grenoble. The energy of the incident X-ray beam was set at 22.0 keV, above the Pb and Se. We observed little to no beam damage at this X-ray energy. We optimized the grazing angle to 0.14° for the best signal-to-noise ratio on both GIWAXS and GIWAXS detectors. The scattering was recorded by two Pilatus detectors. The GIWAXS patterns were recorded on a Pilatus 300 K detector with 619 × 487 pixels, each 172 × 172 μm² in size, positioned approximately 25 cm from the sample. The GISAXS patterns were recorded on a Pilatus 300K-W detector with 1475 × 195 pixels, each 172 × 172 μm² in size, positioned 0.988 m from the sample. Before drop casting the dispersion of NCs on top of the EG substrate, the X-ray beam was aligned to the surface. The XRR patterns were collected on a 1-D Mythen detector. The self-assembly of the NCs was performed in a home-built liquid cell, which can be flushed with nitrogen repeatedly to lower the oxygen and water levels (Supplementary Fig. 21). A Teflon Petri dish (∅ 64 mm) was filled with 28 mL of EG. To EG, we added 100 μL of a 31.7 μM oleic acid solution in EG. The cell was then flushed five times with vacuum/nitrogen cycles. Next, the PbSe NC solution (1 mL; $1.2 \times 10^{-6}$ mol L$^{-1}$ for all solutions) was deposited on top of the liquid substrate. A photograph of the experimental setup, penetration depth calculations and additional GISAXS simulations can be found in Supplementary Figures 21-23.

## Data availability

The datasets generated and/or analyzed during the current study are available from the corresponding author on reasonable request.

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

## Acknowledgements

We are very grateful for the excellent experimental support of Andrey Chumakov, who, besides O.V.K., helped out as local contact during the X-ray scattering experiments at ID10, ESRF. F.M. acknowledges support by the European Comission via the Marie-Sklodowska Curie action Phonsi (H2020-MSCA-ITN-642656). D.V. wishes to thank the Dutch FOM (program DDC13), NWO-CW (Toppunt 718.015.002), and the European Research Council under HORIZON 2020 (Grant 692691 FIRSTSTEP) for financial support. J.J.G. gratefully acknowledges financial support from the joint UU and ESRF Graduate Program.

## Author contributions

J.J.G., E.G., C.v.O., F.M., and M.R.S. conducted the in situ experiments, under supervision of D.V., A.V.P., and O.V.K. G.S. performed the analytical calculations on the adsorption geometries. J.J.G. and E.G. analyzed the GISAXS and GIWAXS data, C.P.H. analyzed the XRR data together with O.V.K. J.J.G. and G.S. wrote the manuscript under supervision of D.V., A.V.P., and O.V.K. with input from all authors.

## Competing interests

The authors declare no competing interests.
