## [Peer Review File · Communications Chemistry]

Reviewers' comments:

Reviewer #1 (Remarks to the Author):

The paper by J. J. Geuchies et al. is possibly a follow up paper of Ref. 11, where an in-situ observation of the formation of PbSe nanocrystal square lattices was reported. The present paper discusses still open questions regarding this square lattice formation. The experimental techniques, a combination of GISAX, GIWAX and XRR is chosen to experimentally find the orientation of the nanocrystals at the liquid/air interfaces, dependent on the nanocrystal size. The experimental findings are fully supported by theoretical predictions, representing a valuable part of the manuscript. The manuscript represents an in-depth investigation of so far missing details, in respect to the formation of nanocrystal monolayers. I find the manuscript highly interesting and very well written. Also the figures are beautiful. Thus I suggest publication without any changes.

Reviewer #2 (Remarks to the Author):

The manuscript by Geuchies et al. describes a combination of X-ray scattering (wide and small angle) and X-ray reflectivity measurements to examine how colloidal nanoparticles assemble at the surface of a liquid interface. The combination of X-ray scattering and X-ray reflectivity is particularly powerful specially to gain combined insights into orientation and vertical position. The experimental results are complemented with theoretical models to describe the thermodynamics of various nanoparticle configurations (i.e., orientation and normal translation relative to the interface.). The authors describe the specific questions of nanoparticle orientation in the more general context of seeking to understand the underlying mechanisms responsible for the formation of superlattice allotropes (e.g., square and honeycomb assemblies) that have been previously reported by their team and others. In lines 60-65, the authors point to key outstanding questions regarding the formation of the allotropes, but the results in the manuscript don't 'close the loop' on the discussion.

The authors present an extensive set of experiments, however, the discussion, arguments and key claims made in the manuscript are not satisfactorily supported by the available evidence. In its current format, the quality of the manuscript falls short of expectations for recommended publication in Nature Communications. Below are some specific comments and suggestions for revision.

The systematic integration of X-ray scattering and X-ray reflectivity is a notable strength of the results presented in this manuscript; the authors show that signals of periodicity can be obtained even from monolayers. The authors compare a range of particle sizes with a size-dependent truncation and show that small nanoparticles orient randomly whereas medium and larger nanoparticles have preferred orientations. This trend can be rationalized considering how the ligand/core size ratio influences the nature of the inter-particle interaction potential. Smaller particles with a larger ligand/core ratio interact as 'softer' spheres whereas the underlying polyhedral shape is more prominent in the medium and larger size particles. The fact that the buckling is only observed ~20% of the time is a bit curious and requires further discussion. What is unique about those 1/5th cases

The observed 'buckling' of the nanoparticle assemblies is arguably one of the most interesting results reported in this manuscript, especially in the context of outstanding unknowns concerning the mechanism by which the related honeycomb structures form. However, they go far in making conclusions based on their data in spite of the low signal to noise ratio (GISAXS), the lack of a $\sqrt{3}$ positioned peak can simply be due to changes in the form factor (simulations needed) or lattice distortions (and related broadening).

Based on the "buckled" structures from XRR data (Fig.4) the authors claim that they measured while there was some toluene left in the film, and then claim that toluene gets trapped in the film, but they don't consider the possibility of local multilayer formation that can stabilize an off-

interface particle position.

The analysis of 'buckled' films could be strengthened significantly by including TEM imaging, as the authors have done in their previous work. TEM analysis would inform whether the film is indeed a buckled monolayer or a multi-layer. Moreover, the buckling described in Fig. 6 could be in various forms, either alternating row displaced above and below, or a (quasi rhombic) checkerboard with neighboring particles displaced above and below.

The other key aspect which isn't yet satisfactorily addressed in this manuscript is the fact that the system of two immiscible liquids (in this case toluene/ethylene glycol /air) contains at least 2 (possibly 3 in the case of incomplete spreading) fluid-fluid interfaces. Depending on the detailed interfacial energies each interface presents an opportunity to form an ordered superlattice. The authors make strong statements on the particles getting trapped in the toluene-air interface ("However, in the experiments, we expect the NC to first adsorb at the toluene/air interface, since the EG/air interface does not exist until all the toluene around a NC is evaporated (and the NCs do not adsorb at the EG/toluene interface, see SI)." and "we assume that the surface tension γ_1 , γ_2 of the NC surface with ethylene glycol and toluene is approximately similar to the surface tension of hexane with ethylene glycol (0.016 N/m at room temperature), and toluene (0 at room temperature, since hexane is miscible in toluene), respectively. At the best of the authors' knowledge, the surface tension between ethylene glycol and toluene at room temperature is $\gamma \approx 0.01$ N/m, implying that $\cos\theta = (\gamma_1 - \gamma_2)/\gamma > 1$. This means that a NC capped by oleic acid does not adsorb at an ethylene glycol/toluene interface and prefers instead to stay in the toluene phase (i.e. without touching the ethylene glycol)"

Russell and co-workers have previously shown that nanoparticles enter into the interface between toluene and water (<https://science.sciencemag.org/content/299/5604/226>). The statement about the relative role of the multiple interfaces in this system is important enough to deserve a separate discussion and perhaps studies on the actual system. Nanoparticles are capped with oleic acid, a molecule with an unsaturated cis bond, and attached to a polar. One cannot reasonably expect that NPs capped with oleic acid will behave the same as hexane in proximity to a nonsolvent that is acidic and hence can desorb ligands increasing the miscibility (ethylene glycol), and 2.). Moreover, the finite solubility of the NPs in toluene vs the infinite miscibility of the hexane-toluene binary system should be a warning sign to make conservative estimates on the stability of a NP system.

The authors show that their observations can be supported by capillary force calculations. However, their arguments is solely based on binding energies at air/EG and air/toluene interface, without performing the calculations at the toluene/EG interface. This is a significant weakness given that the vagueness of the macroscopic thermodynamic estimates.

Finally, the paper is not written in the concise and factual manner expected for a publication in NatComm. Even if all the statements were fully supported, the text could be shorted and made more to the point.

Minor: fix the caption of Fig S19 that refers to Chapter 4 (presumably referring to a thesis).

Reviewer 1

Comment: The paper by J. J. Geuchies et al. is possibly a follow up paper of Ref. 11, where an in-situ observation of the formation of PbSe nanocrystal square lattices was reported. The present paper discusses still open questions regarding this square lattice formation. The experimental techniques, a combination of GISAX, GIWAX and XRR is chosen to experimentally find the orientation of the nanocrystals at the liquid/air interfaces, dependent on the nanocrystal size. The experimental findings are fully supported by theoretical predictions, representing a valuable part of the manuscript. The manuscript represents an in-depth investigation of so far missing details, in respect to the formation of nanocrystal monolayers. I find the manuscript highly interesting and very well written. Also the figures are beautiful. Thus I suggest publication without any changes.

Response: We thank the reviewer for his/her interest and very positive assessment of our work.

Reviewer 2

Comment: The manuscript by Geuchies et al. describes a combination of X-ray scattering (wide and small angle) and X-ray reflectivity measurements to examine how colloidal nanoparticles assemble at the surface of a liquid interface. The combination of X-ray scattering and X-ray reflectivity is particularly powerful specially to gain combined insights into orientation and vertical position. The experimental results are complemented with theoretical models to describe the thermodynamics of various nanoparticle configurations (i.e., orientation and normal translation relative to the interface.).

Response: We thank the reviewer for his/her detailed analysis and careful evaluation of our work, and for the pertinent comments and doubts he/she raised, which we wish to fully address in the revised version of our manuscript.

Comment: The authors describe the specific questions of nanoparticle orientation in the more general context of seeking to understand the underlying mechanisms responsible for the formation of superlattice allotropes (e.g., square and honeycomb assemblies) that have been previously reported by their team and others. In lines 60-65, the authors point to key outstanding questions regarding the formation of the allotropes, but the results in the manuscript don't 'close the loop' on the discussion. The authors present an extensive set of experiments, however, the discussion, arguments and key claims made in the manuscript are not satisfactorily supported by the available evidence. In its current format, the quality of the manuscript falls short of expectations for recommended publication in Nature Communications.

Response: As the reviewer mentions, we did not fully confirm the formation mechanism for the buckled honeycomb superlattices. Still, we provide the first set of in-situ data on the 3D particle adsorption geometry that is an essential piece of information that will contribute to the understanding of the formation mechanism. We added an additional sentence to the introduction to clarify this point.

Action: Page 6, end of introduction: "The data presented here is (also) an important step forward in the understanding of the formation mechanism of two-dimensional superlattices, but the main goal of this work goes beyond this: we unraveled the three-dimensional adsorption profile of PbSe NCs at fluid-fluid interfaces, using a unique and novel combination of different experimental and numerical techniques."

Below are some specific comments and suggestions for revision.

Comment: The systematic integration of X-ray scattering and X-ray reflectivity is a notable strength of the results presented in this manuscript; the authors show that signals of periodicity can be obtained even from monolayers. The authors compare a range of particle sizes with a size-dependent truncation and show that small nanoparticles orient randomly whereas medium and larger nanoparticles have preferred orientations. This trend can be rationalized considering how the ligand/core size ratio influences the nature of the inter-particle interaction potential. Smaller particles with a larger ligand/core ratio interact as 'softer' spheres whereas the underlying polyhedral shape is more prominent in the medium and larger size particles. The fact that the buckling is only observed

~20% of the time is a bit curious and requires further discussion. What is unique about those 1/5th cases?

Response: We thank the reviewer for his continued positive assessment of the experimental techniques presented throughout the paper. His/her reasoning points to great work from the Talapin group [1], and more recently also from the Glotzer group [2], who studied the effect of the ligand/core ratio (as a term of the ‘softness’ of the NCs) on the self-organization of NCs into superlattices.

As for the observation that in roughly 20% of the experiments on the larger particles, we observe a buckled structure; this can be just statistics. If there is an energy barrier in the order of kT , to get adsorbed at the interface, then you will get a probability for adsorption: see Boltzmann factor $e^{-\Delta(E)/kT}$. What we speculate (we do not make any harsh conclusion on this point) in the manuscript is that there is an energy barrier due to the toluene still in the ligand corona. We do think the buckling is a very interesting observation that gives important hints towards the formation of 2D superlattices.

Action: We have added the two beforementioned references to our manuscript and added a sentence to the introduction – page 3: “The exact (binary) superlattice structure which forms depends on interactions between the NCs^{1,2} and the interaction of the particles with the interface”.

- [1] Boles, M. A. & Talapin, D. V. Many-Body Effects in Nanocrystal Superlattices: Departure from Sphere Packing Explains Stability of Binary Phases. *J. Am. Chem. Soc.* **137**, 4494–4502 (2015).
- [2] LaCour, R. A., Adorf, C. S., Dshemuchadse, J. & Glotzer, S. C. Influence of Softness on the Stability of Binary Colloidal Crystals. *ACS Nano* acsnano.9b04274 (2019). doi:10.1021/acsnano.9b04274

Comment: The observed ‘buckling’ of the nanoparticle assemblies is arguably one of the most interesting results reported in this manuscript, especially in the context of outstanding unknowns concerning the mechanism by which the related honeycomb structures form. However, they go far in making conclusions based on their data in spite of the low signal to noise ratio (GISAXS), the lack of a $\sqrt{3}$ positioned peak can simply be due to changes in the form factor (simulations needed) or lattice distortions (and related broadening).

Response: We agree with the reviewer that the buckling of the NC monolayers is an exciting and interesting result that we report. We think this is a good indication that there is toluene left in the ligand corona of the NCs which prevents sinking of NCs into EG. Theoretical calculations based on our estimations for the surface tensions say that all NCs should strongly adsorb at EG/air interface, and not adsorb at EG/toluene interface. We however strongly disagree that we over interpret our data. As a very important note: the buckling is not indirectly deduced from the GISAXS patterns, but directly from the XRR experiments.

The reviewer is correct that through extensive modelling of the GISAXS data, we can also get three-dimensional information. We feel that this is a very indirect approach, especially since we get the direct 3D component of the structure from the XRR data. An example of the influence of structural parameters on the GISAXS pattern was shown in the supporting information of our earlier work. The XRR data is far more direct: the specularly reflected X-ray photons only carry out-of-plane momentum

(i.e. they only have a component in q_z) and hence carry density information about the structure in the z-direction; the direction perpendicular to the liquid interface. The signal-to-noise ratio in the XRR measurements is excellent and our modelling of the patterns is extensive and detailed. This can all be found in the supporting information.

Action: We added a simulated GISAXS pattern for an hexagonal monolayer with and without broadening and changes of angle between the NCs to the supporting information:

Figure S20: Simulations of disorder on the GISAXS pattern of 2D self-assembled nanocrystal lattices. We varied (1) the angle between the nanocrystals, going from an hexagonal to square two-dimensional structure, and (2) the amount of disorder (related to the domain size) in the superlattice, by adjusting the broadening of the structure factor peaks. In the bottom row, we use a relatively small broadening (1%) whereas in the top row, we increased the amount of broadening (to about 6%). Note that **we did not adjust disorder in particle sizes**, since this would decrease the depth of the minima in the form factor.

We also changed a sentence, clarifying that the presence of toluene in the ligand corona is a hypothesis we bring forth - page 20, last sentence: “We hypothesize that a thin layer of toluene was still on top of the EG phase when the X-ray scattering measurements were done,...”

Comment: Based on the “buckled” structures from XRR data (Fig.4) the authors claim that they measured while there was some toluene left in the film, and then claim that toluene gets trapped in the film, but they don’t consider the possibility of local multilayer formation that can stabilize an off-interface particle position.

Response: The reviewer is correct that we speculate on the presence of toluene in the ligand corona of the NCs. Local multilayer formation that stabilizes off-interface particle positions is not the correct explanation in our opinion, since the NCs are oriented 100-up, which is in contrast with our current understanding of the silicene-honeycomb formation, see [Soligno, Vanmaekelbergh, Phys. Rev. X (2019)].

Comment: The analysis of ‘buckled’ films could be strengthened significantly by including TEM imaging, as the authors have done in their previous work. TEM analysis would inform whether the film is indeed a buckled monolayer or a multi-layer. Moreover, the buckling described in Fig. 6 could be in various forms, either alternating row displaced above and below, or a (quasi rhombic) checkerboard with neighboring particles displaced above and below.

Response: We agree that the analysis could be strengthened by additional TEM analysis.

Moreover, analysing the buckled structure of the NC monolayers is not as straightforward as the reviewer presents here. A full tomography study should be done on the NC layers in order to retrieve the three-dimensional, since normal TEM only shows 2D projections of the superlattices (see e.g. our work on the honeycomb superlattice [Boneschanscher et al., Long range orientation and atomic attachment of nanocrystals in 2D honeycomb superlattices, Science 344 (6190), 1377-1380, 2014]). More importantly, we also showed in earlier work that the structures that float on/at the liquid-air interface can be significantly different from the resulting structures that are scooped on a TEM grid due to all sorts of drying effects/forces that act on the particles as the TEM grid is prepared. We feel that the X-ray scattering study gives a clear and in-situ picture of the NC monolayers.

We did take samples after each experiment, but unfortunately, the TEM grids did not survive the trip back from the synchrotron.

Comment: The other key aspect which isn’t yet satisfactorily addressed in this manuscript is the fact that the system of two immiscible liquids (in this case toluene/ethylene glycol /air) contains at least 2 (possibly 3 in the case of incomplete spreading) fluid-fluid interfaces. Depending on the detailed interfacial energies each interface presents an opportunity to form an ordered superlattice. The authors make strong statements on the particles getting trapped in the toluene-air interface (“However, in the experiments, we expect the NC to first adsorb at the toluene/air interface, since the EG/air interface does not exist until all the toluene around a NC is evaporated (and the NCs do not adsorb at the EG/toluene interface, see SI).” and “we assume that the surface tension γ_1 , γ_2 of the NC surface with ethylene glycol and toluene is approximately similar to the surface tension of hexane

with ethylene glycol (0.016 N/m at room temperature), and toluene (0 at room temperature, since hexane is miscible in toluene), respectively. At the best of the authors' knowledge, the surface tension between ethylene glycol and toluene at room temperature is $\gamma \approx 0.01$ N/m, implying that $\cos\theta = (\gamma_1 - \gamma_2)/\gamma > 1$. This means that a NC capped by oleic acid does not adsorb at an ethylene glycol/toluene interface and prefers instead to stay in the toluene phase (i.e. without touching the ethylene glycol)".

Response: We thank the reviewer for this comment.

1. The EG/air interface is not accessible to the NCs, even in the case of toluene partial wetting on the EG substrate, until complete evaporation of the solvent, since the NCs cannot fully desorb from the solvent (the energy penalty would be too great, of the order of 10-to-100 kT, see our calculations).
2. The NCs should not adsorb at the EG/toluene interface since **we are in the regime of complete wetting of toluene on the NC's surface**, based on our surface tension estimations: the surface tension hexane/EG is 0.016 N/m [1], while the surface tension toluene/EG should be lower than 0.010 N/m [2].

We agree that the surface tension of the NC's surface capped by oleic acid with the surrounding medium is not necessarily equal to the surface tension of hexane with the same medium, but these two surface tensions should be reasonably similar. From our text it is clear that this is just a (reasonable) estimation, which we do since more precise data are not available.

[1] G. ZOGRAFI and S. H. YALKOWSKY, *Journal of Pharmaceutical Sciences* **63**, 1533 (1974).

[2] R. Aveyard *et al.*, *Langmuir* **9**, 523 (1993).

Action:

To page 36 of the supporting information, we added the additional sentence: "We would like to point out that the surface tensions we use are reasonable estimates, since, to our knowledge, there is no better experimental data or theoretical prediction available. We reasonably expect that the surface tension of the oleate-capped NC's surface with the surrounding medium is similar the surface tension of hexane with the same medium."

Comment: Russell and co-workers have previously shown that nanoparticles enter into the interface between toluene and water (<https://science.sciencemag.org/content/299/5604/226>).

Response: See our response to point above. We have also added the reference to the introduction.

Comment: The statement about the relative role of the multiple interfaces in this system is important enough to deserve a separate discussion and perhaps studies on the actual system

Response: We agree with this comment of the reviewer. As mentioned on page 17, end of the first paragraph, we plan to study multiple interfaces in the future. For this we need a specially designed cell which allows for very well controlled evaporation of the toluene.

Comment: Nanoparticles are capped with oleic acid, a molecule with an unsaturated cis bond, and attached to a polar. One cannot reasonably expect that NPs capped with oleic acid will behave the same as hexane in proximity to a nonsolvent that is acidic and hence can desorb ligands increasing the miscibility (ethylene glycol),

Response:

1. As already illustrated in our response to one of the reviewer's points above, we deem reasonable to approximate that the surface tension of the NC's surface covered by oleic acid is similar to the surface tension of hexane, as a first-order approximation, since only the oleic acid tails will be in contact with the surrounding medium.
2. The presence of the unsaturated cis bond should not drastically affect the behaviour of the NC surface covered by oleic acid with respect to its surface tension: for example, the surface tension hexane/air, which is 0.018 N/m at room temperature, is the same of the surface tension of 1-hexene/air at room temperature (see e.g. <https://materials.springer.com/bp/docs/978-3-540-75508-1>).
3. We agree with the reviewer that ethylene glycol can indeed induce desorption of ligands from NC surface, and this is, we believe, one of the chemical mechanisms at the base of the honeycomb formation, see Soligno and Vanmaekelbergh, *Phys. Rev. X* **9**, 021015 (2019). In the before mentioned reference, we keep into account that at some point ligands desorb by considering a lower contact angle for the NC {100} facets, i.e. where the ligands are expected first to detach, since these facets will decrease their chemical affinity with the solvent when the ligands detach. In this work, however, we believe ligand detachment from the NCs' facets is prevented since we added oleic acid to the system [see page7, second paragraph, "we add 100 μ L of a 31.7 μ M oleic acid solution in ethylene glycol..."]. The fact that ligands did not desorb from the NCs' surface is proved by the fact that we did not observe (almost) any oriented attachment event, as we mention in the manuscript on page 13 of the main text, where we discuss the GIWAXS analyses.

Comment: and 2.). Moreover, the finite solubility of the NPs in toluene vs the infinite miscibility of the hexane-toluene binary system should be a warning sign to make conservative estimates on the stability of a NP system.

Response: We deem important to point out that the finite solubility of NCs in toluene is (also) due to the finite size of the NCs and NC-NC hard interactions: at high packing fraction, the NCs will crystallize. Of course, we remark once more, in agreement with the reviewer's remarks, that our estimations for the surface tension of NCs capped by oleic acid are only approximations (used as more precise data on these surface tensions is currently not available) and it is not clear how precise these are.

Comment: The authors show that their observations can be supported by capillary force calculations. However, their arguments is solely based on binding energies at air/EG and air/toluene interface, without performing the calculations at the toluene/EG interface. This is a significant weakness given that the vagueness of the macroscopic thermodynamic estimates.

Response: Again, we point out that our NCs are not expected to adsorb at the EG/toluene interface since they are covered by oleate ligands, hence they are in the complete wetting regime at this interface (see our argument based on the surface tensions illustrate above).

Action: See points above.

Comment: Finally, the paper is not written in the concise and factual manner expected for a publication in NatComm. Even if all the statements were fully supported, the text could be shorted and made more to the point.

Response: As a minor point, the paper is submitted to Communications Chemistry (not Nat. Comms.). We disagree that the paper is not written in a concise and factual manner; we based our discussion on the experimental results and tried to rationalize these using theoretical calculations of the adsorption geometry. We also think the text is written in such a way that people without prior knowledge to the scattering and reflectivity techniques we present can fully follow the line of thinking. For these reasons we do not agree that shortening the text is for the best interest of the readability and quality of the manuscript.

Comment: Minor: fix the caption of Fig S19 that refers to Chapter 4 (presumably referring to a thesis).

Response: We thank the reviewer for his very thorough reading of the manuscript and for noting this typo; we have removed it from the caption of Figure S19.

REVIEWERS' COMMENTS:

Reviewer #2 (Remarks to the Author):

The authors have addressed all of the comments and suggestions made in the initial review of the manuscript. The manuscript is recommended for publication.